# Factors influencing infertile couples' decisions to seek healthcare: A mixed methods study from Islamabad, Pakistan

**Menahyl Mahmood, Babar Tasneem Shaikh⬤, Mariam Ashraf⬤***

School of Public Health, Health Services Academy, Islamabad, Pakistan

* maryamashraf@hotmail.com

## Abstract

This study examines the factors influencing the decision-making of infertile couples to seek healthcare, focusing on socio-demographic characteristics and barriers to accessing care. The study uses a mixed-methods approach, including in-depth qualitative interviews (IDIs) and a cross-sectional survey of women experiencing infertility. A hospital-based cross-sectional study was conducted from July to December 2024, including 269 participants, along with in-depth interviews with 12 women seeking infertility treatment. Qualitative findings showed significant emotional distress among participants, with many feeling judged and inadequate. Social isolation and external stigma were common, worsening the emotional burden. Limited access and inconsistent support further increased the difficulties faced by infertile couples. Of the 269 participants, 43.9% reported primary infertility, and 56.1% secondary infertility. Most had been married for 1–5 years and had been delaying treatment for over three years. The main reason for seeking medical treatment was the inability to conceive, followed by family pressure. Financial constraints, lack of access, and social stigma were major barriers to healthcare. Regression analysis indicated that individuals with lower educational levels were less likely to seek healthcare than those with higher levels. Couples in joint families were more inclined to pursue treatment, and primary infertility was a key factor influencing the decision to seek medical care. Community-based awareness programmes are vital to dispel myths surrounding infertility, alongside counselling services to address emotional stress. Healthcare providers should receive training to offer empathetic support tailored to infertile couples. Furthermore, social health protection schemes should consider subsidized treatment or insurance coverage to alleviate financial burdens for low-income families in Pakistan.

## Introduction

Globally, infertility is a neglected public health issue that often leads to marital discord [1]. Childlessness is often used interchangeably with infertility or subfertility. Infertility

**Data availability statement:** All data and related metadata underlying the findings are reported in a submitted manuscript. However data is submitted in figshare repository and following is the link https://doi.org/10.6084/m9.figshare.31150813.

**Funding:** The authors received no specific funding for this work.

**Competing interests:** The authors have declared that no competing interests exist.

is defined as "the inability to achieve pregnancy within a specified period, typically more than one year, among couples of reproductive age engaging in regular, unprotected sexual intercourse" [2]. Infertility may be classified as either primary or secondary. Primary infertility refers to couples who have never achieved pregnancy. In contrast, secondary infertility pertains to those who have experienced at least one previous pregnancy, regardless of its outcome (live birth, stillbirth, miscarriage, ectopic pregnancy, or induced abortion) [3].

For centuries, individuals have relied on traditional practices to address fertility challenges. These practices continue to vary widely across cultures today. Common methods used worldwide include visiting religious leaders or sacred sites, rituals involving animals (such as feeding a red rooster to a child), wearing amulets, drinking zamzam water (from Mecca), and utilising herbs, mineral supplements, chiropractic care, naturopathy, homoeopathy, and acupuncture [4,5]. Since fertility problems are often associated with women, traditional practices are more commonly used in women. However, fertility is a problem that affects couples [6]. Health professionals working in this domain should be aware of the traditional practices used by couples, the benefits and harms of practices, and should give proper counselling. In addition, determining the learning paths of practices will provide data to guide couples toward the right resources [7].

The female infertility rate in Pakistan is approximately 22%, with 4% experiencing primary infertility and 18% dealing with secondary infertility. When examining infertility in the Pakistani context, it is essential to consider Islamic laws and cultural customs. A married woman is permitted to try for a child within two or more years of marriage without using contraceptives. In our society, infertility is viewed not only as a biological issue but also as a social stigma, as cultural attitudes, norms, and religious beliefs often associate infertility with personal, interpersonal, or societal failure. Additionally, in developing Asian countries, infertility is often seen as an act of God, a punishment for past sins, or the result of witchcraft [8].

These cultural beliefs significantly influence healthcare-seeking decisions among infertile couples. Beliefs in evil forces and supernatural powers as causes of infertility remain prevalent, particularly among populations with lower educational levels, leading couples to seek treatment from faith healers before consulting medical professionals [9,10]. Research from Pakistan demonstrates that 35.2%-44.2% of individuals consider spiritual healers as a primary treatment option for infertility [11]. This perception varies notably across educational levels and socioeconomic strata. Studies from India reveal that couples with higher education (more than 10 years of schooling) are three times more likely to seek allopathic treatment compared to those with no formal education [10,12]. Conversely, couples with lower education levels may delay medical consultation by 1–3 years, instead pursuing spiritual or traditional solutions [12,13]. Economic factors compound this issue, as traditional remedies are often perceived as more affordable than medical treatment, creating a complex interplay between cultural beliefs and financial constraints that shapes treatment-seeking timelines [14,15]. In South Asian contexts, where infertility is frequently attributed to divine punishment or ancestral curses, couples often prioritize religious rituals and

traditional sacrifices over biomedical interventions, particularly in rural areas where healthcare literacy remains limited [7,16].

In South Asia, social stigma is associated with infertility and hence results in a higher prevalence of physical violence against infertile women. These women often face inheritance deprivation, physical abuse, ostracization, exile to their parental homes, marriage dissolution, and verbal abuse [10,11]. The emotional strain of infertility can lead to a range of psychological disorders, including hostility, anxiety, frustration, self-blame, and even suicidal thoughts [17]. Data suggests that by age 38, the cumulative incidence of infertility ranges from 14.4% to 21.8% for men and from 15.2% to 26.0% for women [18].

In Pakistan, family life and values are deeply ingrained in society. The traditional family unit is sustained by the agreement that men and women will fulfil their mutually exclusive yet complementary gender roles within the household [19]. The ideal family structure is a joint family, with a male-headed household consisting of the male head, his wife, their adult sons with their wives, and their children, all cohabiting harmoniously. The preferred number of children is four, with at least two sons. Families are commonly formed through endogamous marriages, often between first or second cousins. Upon marriage, the bride takes up residence in her husband's family home, where she occupies a structurally weak position, subordinate not only to the men but also to the senior women in the household, particularly the mother-in-law [20].

This study investigates the various factors influencing how married couples decide whether to seek medical care for infertility, including social, psychological, economic, and cultural considerations.

## Study research objectives and questions

This research study aimed to answer three research questions. The first two questions were assessed through a quantitative survey, and the third one was evaluated through qualitative information

Research Objectives: The objectives of this research are to

1. Explore the influence of social and cultural norms on the healthcare-seeking behaviour of infertile couples.

2. Assess the factors affecting the decision to seek medical treatment for infertility.

3. Understand the issues and challenges faced by women seeking infertility treatment.

### Research questions

1. How do social and cultural norms influence the healthcare-seeking behavior of infertile couples?

2. What factors most significantly affect the decision-making process for infertile couples when seeking medical treatment?

3. What are the primary issues and challenges faced by women in the pursuit of infertility treatment?

## Materials and methods

### Ethics statement

The research was conducted in accordance with the principles outlined in the Declaration of Helsinki. All participants were informed of the research aims and objectives. Informed consent was obtained from each participant through verbal consent, and where possible, written consent was sought to ensure that all participants fully understood that the data collection is solely for research purposes.

The measures for collecting information, including protocols to maintain anonymity and confidentiality, were strictly followed. No names or identifiable details were recorded, and participants were de-identified in the final dataset to protect

their identities. Participants were informed that they could refuse or withdraw from the study at any time without providing a reason. Ethical approval was granted by the Institutional Review Board at the Health Services Academy, Islamabad, through letter F.No. 000624/HSA-MSPH/2023.

**Study design, participants, and sampling strategy**

The research employed a mixed-methods approach, combining qualitative and quantitative data collection methods. The most recognised method for integrating these approaches is the Convergent study design [21], which aims "to obtain different but complementary data on the same topic to understand the research problem. " This design seeks to leverage the strengths of quantitative methods while addressing their limitations. It is a one-phase approach where both qualitative and quantitative data are collected simultaneously, earning it the label "concurrent design. " Typically, it involves collecting and analysing qualitative and quantitative data separately but concurrently to gain a comprehensive understanding. The study used the convergence model (Fig 1), representing the traditional triangulation approach, where data is collected and analysed independently, then compared during interpretation to synthesize findings.

The quantitative component consisted of a facility-based cross-sectional survey, while the qualitative part included in-depth interviews (IDIs) with infertile women seeking treatment at the facility. The study population comprised married couples in Islamabad who had unsuccessfully tried to conceive for 12–18 months, experiencing primary and secondary infertility (having at least one live child), aged 18–35 years and were recruited from a private hospital providing infertility treatment. The sample size was calculated using OpenEpi, based on a 22% prevalence rate of infertility in Pakistan [8], which yielded a required sample of 268 couples at a 95% confidence interval with a 5% margin of error. Recruitment followed strict inclusion and exclusion criteria to ensure relevance. For the qualitative component, twelve in-depth interviews were conducted with women facing primary and secondary infertility using purposive sampling to capture diverse experiences; recruitment continued until thematic saturation was achieved. The decision to include only women in the qualitative component was deliberate and grounded in the study's third research objective, which specifically sought to understand the issues and challenges faced by women during the treatment-seeking process. Existing literature from South Asia consistently shows that the social and emotional burden of infertility falls disproportionately on women, who are more frequently blamed, isolated, and subjected to pressure from family members [9,10]. A women-focused qualitative inquiry was therefore considered the most appropriate approach to elicit these experiences in depth. We acknowledge, however, that excluding male partners limits insight into men's perspectives, and future research should incorporate qualitative data from both partners to provide a more complete picture Alongside the survey, twelve IDIs were conducted with women facing primary as well as secondary infertility to explore the perceptions about the issues and challenges they had to face while seeking treatment.

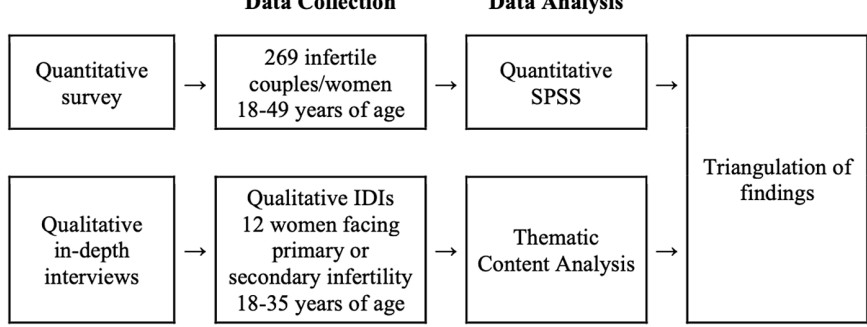

**Fig 1. Schematic Figure showing mix-method approach for this research.**

## Inclusion and exclusion criteria for qualitative and quantitative surveys

### Inclusion criteria.

• Couples/women who were facing primary infertility and have failed to conceive for ≥12 months or more

• Couples/women facing secondary infertility and have not been able to conceive for ≥12 months or more

• Neither partner currently using contraception

• Not currently pregnant or lactating

• Seeking or having sought infertility treatment

• All those who consented to participate in the research study were interviewed

### Exclusion criteria.

• Participants with severe comorbid conditions.

Based on the above criteria, the response rate was 100% with no refusals.

## Survey tool and guidelines

A closed-ended questionnaire was used to assess knowledge, attitudes, and practices related to infertility. The data collection instrument was an adapted and modified version of a validated survey from a study conducted in Mumbai, India, to ensure comprehensive insights [13]. The questionnaire consisted of four sections (A, B, C, and D); Section A included questions on socio-demographic characteristics. Section B contained questions related to infertility. Section C gathered information about couples' attitudes towards infertility. Section D included items related to decision-making for seeking care. (S1 File). The reliability of the instrument was determined using the test-retest method. The tool was pilot-tested on 15 participants at an infertility clinic different from the original research site. Changes were made accordingly, and the test was re-administered two weeks later. A reliability coefficient of 0.85 was obtained using Cronbach's alpha.

For qualitative data, open-ended guidelines were developed for women facing challenges pertinent to infertility.

## Data collection, management and analysis

Survey data were gathered from 10 July to 30 December 2024 through face-to-face interviews using consecutive sampling. All eligible couples/women presenting to the infertility clinic during the study period who met the inclusion criteria and provided informed consent were enrolled until the required sample size was achieved. This approach ensured comprehensive capture of the study population accessing care during the specified timeframe. The survey tool was initially prepared in English. To ensure it was understandable to local participants, an independent bilingual expert translated it into Urdu, maintaining neutrality and accuracy. A second independent bilingual individual then back-translated the Urdu version into English. This back-translated version was carefully compared with the original to verify consistency, and any discrepancies identified during this process were thoughtfully corrected to ensure clarity and reliability in the final version.

The IDIs were conducted with women experiencing infertility and continued until data saturation was reached. Participants were chosen based on their willingness to share detailed insights into their experiences with infertility. The interviews were conducted in Urdu, recorded, and transcribed for analysis. The written and recorded materials were then translated into English. Field researchers with experience in both data collection and data entry were recruited and trained, and a quality assurance officer was assigned to ensure data quality.

Quantitative data were analyzed using SPSS version 24. First, data cleaning and validation procedures were performed to identify missing values, outliers, and data entry errors. Descriptive statistics, including frequencies, percentages,

means, and standard deviations, were calculated to summarize demographic characteristics, infertility profiles, and care-seeking patterns.

For bivariate analysis, chi-square tests (or Fisher's exact test when cell counts were <5) assessed associations between the primary outcome variable (decision to seek medical care) and independent variables including sociodemographic factors (age, education, income, family structure), infertility characteristics (primary vs. secondary, duration), and psychosocial factors (family pressure, stigma perception, decision-making autonomy).

Variables demonstrating associations with $p < 0.20$ in univariate analysis were considered for inclusion in the multivariate model to avoid premature exclusion of potentially relevant predictors. A binary logistic regression model using the enter method was constructed with "decision to seek medical care" as the dependent variable. The final parsimonious model retained only statistically significant predictors ($p < 0.05$). Model assumptions were verified: multicollinearity was assessed using Variance Inflation Factors (VIFs < 5 for all variables), and the Hosmer-Lemeshow test confirmed adequate model fit ($\chi^2 = 3.42$, df = 8, $p = 0.80$). Adjusted odds ratios (OR) with 95% confidence intervals (CI) were calculated to quantify associations, controlling for potential confounders

For qualitative data, interview notes were documented, and interviews were recorded whenever possible, with verbatim transcripts produced. Qualitative data analysis started simultaneously with data collection and was an iterative, ongoing process. Thematic content analysis was carried out manually using an inductive approach. Two researchers independently coded the transcripts line by line, generating initial codes from early interviews. These codes were gradually organised into broader themes and subthemes through iterative comparison across transcripts. A codebook was developed collaboratively and refined as coding progressed. Any discrepancies in coding between the two researchers were resolved through discussion and consensus before finalising the thematic framework. Data saturation was monitored throughout the data collection process and was considered reached when no new themes or subthemes emerged from successive interviews, which occurred at the twelfth interview.

Data triangulation was employed to ensure validity and reliability. Triangulation involved cross-checking results from three sources: survey data, interview transcripts, and published literature. Quantitative patterns, such as the link between family structure and care-seeking, were compared with qualitative themes, such as social pressure from joint family members, to identify areas of agreement or discrepancy. When both sources aligned, it reinforced the credibility of the findings. In cases of divergence, an interpretive analysis provided a deeper, contextually grounded explanation. The application of this approach to data collection and analysis enhances the credibility and robustness of the study's findings.

## Results

### Qualitative findings

Qualitative findings from the IDIs emphasized the emotional and social challenges faced by women experiencing infertility. Of the twelve women interviewed, seven sought treatment for primary infertility and five for secondary infertility. More than half of the women (n = 7) were visiting the facility for the first time. Most had been trying to conceive without any medical intervention for the past 18–36 months. The majority of women consistently reported experiencing intense emotional distress. Many described feelings of judgment and inadequacy, feeling scrutinised even while seeking solutions. They shared that the infertility treatment process is emotionally exhausting, characterised by cycles of hope and disappointment that increase anxiety and depression.

*'Seeking infertility treatment is tough; you are still unsure of the outcome.'*

### Women seeking infertility treatment

A recurring theme was social withdrawal and feelings of loneliness. Participants described feeling isolated due to constant questioning from family members and friends, which worsened their sense of loneliness and alienation.

Participants felt compelled to meet societal expectations about fertility and motherhood, which intensified their stress.

> *'Immediately after marriage, we are constantly asked about pregnancy why aren't you getting pregnant? This question itself is a major worry.'*

### Women seeking infertility treatment

All participants reported that the most common barrier they face in treating infertility is finding the right treatment. Public facilities are overwhelmed, and private facilities are costly. As a result, choosing the right facility and healthcare provider becomes difficult. Additionally, couples often face high costs for infertility treatments, with many insurance plans not covering these expenses, which increases their emotional strain. In some regions, women struggle to access fertility specialists and advanced treatments, often needing to travel long distances or accept less effective options.

> *Apart from the initial medical treatment that we receive, all other medical tests and treatments are very expensive and they consume time as well as money.*

### Women seeking infertility treatment

> 'Insurance only covers normal cases, but non of the insurance plans cover infertility treatments.'

### Women seeking infertility treatment

Some women reported feeling that their concerns are not adequately addressed or taken seriously, which can lead to feelings of frustration and helplessness throughout their treatment journey. This lack of emotional support may also impact their overall experience and potentially influence treatment outcomes.

Women reported that the persistent pressure exerted by their spouse, as well as their parents or parents-in-law, played a significant role in influencing their decision to seek medical treatment for infertility. The women shared that in our society, it is the woman who is blamed for not conceiving, and little concern is raised regarding men's health when it comes to becoming a father.

### Quantitative findings

**Sociodemographic characteristics.** The study respondents primarily consisted of 240 (89.2%) couples and 29 (10.8%) women visiting the healthcare facility. The mean age of participants was 28.4 years (SD = 4.7), with 168 (62.5%) falling within the 20–34 years age range.Most women were housewives (205, 76.2%), lived in joint families (185, 68.8%), and 137 (50.9%) had secondary education. The majority belonged to the middle class, numbering 170 (63.2%), while only 4 (1.5%) were from high-income backgrounds.

Among 269 couples, 118 (43.9%) reported primary infertility and 151 (56.1%) secondary infertility. Most had been married for 1–5 years (n = 130, 48.3%), and 120 (44.6%) had been attempting to conceive for the past 1–3 years. About 178 (66.2%) had previously sought medical advice, with 122(45.4%) delaying treatment for over three years after marriage. Table 1

### Health-seeking behaviour

As shown in Table 1, most participants (177, 65.8%) initially consulted obstetricians/gynaecologists, while only 3 (1.1%) consulted fertility specialists.. A self-motivated desire for parenthood was the primary driver of care-seeking, reported by 120 participants (44%), Inability to conceive, reported by 120 (44%), was the primary reason for seeking medical treatment, followed by family pressure, reported by 78 (29%). Financial barriers, stigma, and access to services appear to contribute

**Table 1. Factors affecting care seeking for infertility.**

| First person consulted | Frequency (n = 269) | Percentage |
|---|---|---|
| Ob/Gyn | 177 | 65.8% |
| Fertility specialist | 3 | 1.1% |
| Friends/family | 44 | 16.4% |
| General practitioner | 45 | 16.7% |
| **Factors influencing the decision to seek medical treatment for infertility** | | |
| Age-related concern | 44 | 16.4% |
| Family pressure | 78 | 29.0% |
| Healthcare professional | 27 | 10.0% |
| Desire for parenthood | 120 | 44.0% |
| **Barriers or Challenges** | | |
| Financial constraints | 90 | 33.4% |
| Lack of access | 80 | 29.7% |
| Social stigma | 99 | 36.8% |
| **Treatment advised by** | | |
| Friends | 23 | 8% |
| Household members | 64 | 23.7% |
| Self-seeking | 176 | 65.4% |
| None* | 6 | 2.2% |
| **Decision to seek care was taken by** | | |
| Both | 210 | 78.1% |
| Husband | 17 | 6.3% |
| Parents | 23 | 8.6% |
| Wife | 19 | 7.1% |
| **Number of consultations** | | |
| 1 | 137 | 50.9% |
| 2-4 | 72 | 26.8% |
| 5-6 | 41 | 15.2% |
| 7 and more | 19 | 7.1% |

***None** refers to participants who sought treatment without any external advice or recommendation from family, friends, or household members. These individuals made independent decisions to seek care without consultation.*

equally to the severity of the issue. Joint spousal decision-making for seeking care was reported by 210 (78.1%), followed by parental influence at 23 (8.6%). Most couples (137, 50.9%) indicated that their first visit was for treatment.

## Factors affecting the decision to seek medical treatment for infertility

The primary outcome of this study was the decision to seek medical treatment for infertility. A chi-square test was conducted to examine the association of various characteristics with the outcome variable. A significant association (Table 2) was found between the type of family system, education level, and treatment advised by (p < 0.05).

Multivariate logistic regression analysis (Table 3) was conducted to identify the predictors for seeking infertility treatment, using a parsimonious model-building approach and screening criteria of p-value <0.20 at the univariate level and <0.05 at the multivariate level. The model was tested for goodness of fit with the Hosmer and Lemeshow test. The final multivariate model was found to be a good fit (p-value <0.80).

**Table 2.  Association of factors influencing in making decision to seek medical care for infertility.**

| Characteristics | | Influence in making Decision for seeking infertility treatment | | *p-value* |
|---|---|---|---|---|
| | | **Yes**<br>**No (%)** | **No**<br>**No (%)** | |
| **Type of family** | Joint | 112 (41.6%) | 73(27.1%) | 0.021 |
| | Nuclear | 63 (23.4%) | 21(7.8%) | |
| **Education level** | Primary/lower | 23(8.5%) | 37(13.7%) | 0.000 |
| | Middle/higher | 152 (56.5%) | 57(21.2%) | |
| **Treatment advised by** | Self-seeking | 107(39.7%) | 75(27.8%) | 0.002 |
| | Family/friends | 68(25.2%) | 19(7%) | |

Chi-square value of <0.05 is considered statistically significant.

**Table 3.  Logistic regression model identifying factors that influence the decision to seek medical treatment for infertility.**

| Characteristic | Category | β | OR - Adj | 95% CI of OR | p-value |
|---|---|---|---|---|---|
| Educational level | Primary and lower | -1.348 | 0.260 | (0.135-1.498) | 0.000 |
| | Middle/higher | 1 | | | |
| Type of family | Joint | 1.039 | 0.354 | (0.183-01.685) | 0.002 |
| | Nuclear | 1 | | | |
| Primary infertility | Yes | 1.180 | 3.254 | (0.546- 6.850) | 0.002 |
| | No | 1 | | | |

* Hosmer and Lemeshow Goodness of Fit test p-value<0.80.

Regression analysis shows that individuals with primary or lower education are 0.2 times less likely to seek healthcare than those with middle or higher education. Similarly, couples living in a joint family system are 0.35 times more likely to seek healthcare for infertility treatment compared to their counterparts. Couples experiencing primary infertility are more inclined to seek medical care compared to their counterparts. It should be noted that the confidence intervals for the family structure variable (95% CI: 0.183–1.685) and the primary infertility variable (95% CI: 0.546–6.850) are relatively wide, suggesting these associations should be interpreted with some caution. While statistical significance was achieved, the possibility of chance findings cannot be fully excluded given the sample size and single-site design. These findings are best viewed as exploratory and warrant replication in larger, multi-site studies.

## Discussion

Infertility treatment-seeking behaviours reflect a complex interplay of sociocultural norms, psychological stress, and systemic barriers [13]. Women bear the brunt, reflecting societal norms holding women responsible for infertility- a trend particularly seen in South Asia as well as in Africa [22], where cultural expectations and stigma force women to seek treatment earlier. Male infertility remains under-addressed, with delays observed in India and China, where men's reluctance stems from fear of challenging norms linking fertility to masculinity [23,24]. Nonetheless, delays in treatment initiation are common for both genders. In our study, many women sought treatment after 3 years of marriage, which concurs with the findings of an Indian study on care seeking in primary infertility [12]. In Rwanda, poor literacy and awareness further prolongs care-seeking [22], while factors like age, self-employment, and longer marriages contribute to delays in Nigeria [25]. These finding also corroborates our results. Psychological barriers such as stigma and denial, further deter timely interventions [26]. The psychological toll is profound and gendered. In our study, women reported isolation, distress and lack of social support, most of the time relying on partners for decisions. Similar emotional burdens are documented in other

studies where stress, self-blame, and depression are seen as common factors [27,28]. Unlike women, men often avoid sharing their struggles, turning to work instead, highlighting the need for integrated psychological care [29].

The strong association between education level and healthcare-seeking behavior identified in our regression analysis (OR=0.260, p<0.001) warrants deeper examination of underlying mechanisms. Lower educational attainment influences treatment-seeking through multiple pathways: (i) **Knowledge gaps**: individuals with limited education often lack understanding of reproductive biology, fertility windows, and available treatment options, leading to delayed recognition of infertility as a medical condition requiring intervention; (ii) **Health literacy barriers**: difficulty navigating complex healthcare systems, understanding medical terminology, and interpreting treatment options creates additional obstacles; (iii) **Economic constraints**: lower education typically correlates with reduced earning capacity, making costly infertility treatments financially prohibitive; (iv) **Social factors**: less educated individuals may have stronger adherence to traditional beliefs about infertility causation (divine punishment, witchcraft) and greater reliance on alternative medicine or spiritual interventions. Furthermore, lower education often intersects with limited social capital and reduced access to information networks that could facilitate appropriate healthcare-seeking. These findings align with studies from India and Rwanda showing that literacy and awareness significantly impact treatment initiation timelines [12,22]. It is important to note, however, that educational attainment does not uniformly determine health-seeking behaviour. In settings such as Pakistan, highly educated and otherwise well-informed individuals may continue to adhere strongly to deeply held generational and cultural beliefs about infertility, including perceptions of divine will or social duty. Education may reduce certain knowledge barriers while leaving cultural and normative influences largely intact. This nuance should be considered when designing education-based interventions, ensuring they address cultural framing alongside information provision.

Cultural beliefs heavily influence treatment pathways. In our study, women turned to alternative medicine, a very common trend in South Asia and the Middle East, where spiritual healers and traditional remedies often precede medical care. Misconceptions, ranging from medical to supernatural causes perpetuate delays, compounded by other narratives describing destiny and evil spirits as causes of infertility [11,30]. Further adding to this state of affairs, financial constraints are another major barrier in care seeking, documented in our study, as well as in other studies from other regions [14,15]. There is a need to design and implement community-based awareness programs to dispel myths and misconceptions about infertility. It is important to include counselling services as part of infertility treatment to provide support for emotional and psychological stress. Healthcare providers must be trained to provide empathetic support tailored to the unique needs of infertile couples. The social health protection program should consider offering subsidized treatment or insurance coverage for infertility treatments to alleviate financial constraints faced by many poor families in Pakistan.

Our logistic regression analysis identified three significant independent predictors of healthcare-seeking behavior for infertility treatment, providing crucial insights for intervention design.

**Educational Attainment**: The strongest predictor was education level (OR=0.260, 95% CI: 0.135-1.498, p<0.001). Individuals with primary or lower education were 74% less likely to seek healthcare compared to those with middle or higher education. This finding aligns with existing literature from India and Rwanda [12,22] demonstrating that education enhances health literacy, reduces stigma, and facilitates navigation of healthcare systems. Educational interventions targeting low-literacy populations through community health workers and visual communication tools could substantially improve care-seeking rates.

**Family Structure**: Living in a joint family system significantly increased the likelihood of seeking treatment (OR=0.354, 95% CI: 0.183-1.685, p=0.002). While counterintuitive given increased family pressure described in qualitative findings, this likely reflects financial pooling and collective decision-making in joint families. Extended family members may contribute financially to treatment costs and provide emotional support, offsetting the negative impacts of stigma and pressure. However, this also suggests that couples in nuclear families face disproportionate barriers and may benefit from community support structures and financial assistance programs.

**Type of Infertility**: Couples experiencing primary infertility were significantly more likely to seek care (OR=3.254, 95% CI: 0.546-6.850, p = 0.002) compared to those with secondary infertility. This finding reflects greater urgency and social pressure for childless couples. Secondary infertility may be perceived as less severe, particularly if couples already have male children, leading to delayed or foregone treatment. Healthcare messaging should emphasize that secondary infertility warrants equal medical attention and that family completion is a legitimate reproductive health goal.

## Conclusion and way forward

Community-based awareness programmes are vital to dispel myths surrounding infertility, alongside counselling services to address emotional stress. Healthcare providers should receive training to offer empathetic support tailored to infertile couples. Furthermore, social health protection schemes should consider subsidised treatment or insurance coverage to alleviate financial burdens for low-income families in Pakistan

Based on our findings, we recommend the following specific policy interventions:

Launch nationwide public awareness campaigns through mass media, mosques, and community centers to normalize infertility as a medical condition rather than a source of shame. Integrate reproductive health education into school curricula and pre-marital counseling programs. Establish peer support groups and online forums for infertile couples to reduce isolation. Train religious leaders and community influencers to deliver accurate, destigmatizing messages about infertility.

Expand the Sehat Sahulat Program, the national health insurance, to include comprehensive infertility diagnostics and treatment coverage. Establish subsidized fertility clinics in public hospitals with sliding-scale fees based on family income. Mandate private insurance plans to cover at least basic infertility investigations and first-line treatments. Additionally, create government-subsidized loan programs with flexible repayment terms to assist with advanced reproductive technologies.

Establish infertility specialty units in public tertiary hospitals across all provinces. Train general practitioners and gynecologists in basic infertility management to provide care in underserved areas. Develop telemedicine consultation services for couples in rural or remote locations. Create referral pathways from primary care to specialized fertility centers.

Mandate psychological counseling as a standard part of all infertility treatment protocols and train healthcare providers in trauma-informed, culturally sensitive counseling techniques. Establish dedicated mental health services within fertility clinics and develop culturally adapted support materials in Urdu and regional languages.. Before any of the above recommendations are implemented at scale, it is strongly advised that Key Informant Interviews (KIIs) be conducted with key stakeholders, including health policymakers, facility administrators, community leaders, and healthcare providers. This would allow for a pragmatic assessment of the feasibility of each recommendation, identification of contextual barriers, and refinement of intervention strategies to ensure they are sustainable and culturally appropriate within the Pakistani health system context.

## Strengths and limitations of this research

The study has several limitations. Firstly, the sample was limited to couples seeking treatment at a single private hospital in Islamabad, which significantly affects the generalizability of findings. This sampling approach introduces several biases: (a) it excludes couples who cannot afford private healthcare and rely solely on public facilities, where services may be limited or unavailable; (b) it does not capture the experiences of couples in rural areas who face geographic barriers to specialized infertility care; (c) private hospital patients likely represent a more economically advantaged subset with higher health literacy and greater agency in decision-making. Consequently, our findings may underestimate the barriers faced by lower-income couples and overrepresent treatment-seeking behaviors among more educated, urban populations. Future research should employ multi-site sampling strategies encompassing both public and private facilities across urban and rural settings to enhance representativeness and capture the full spectrum of healthcare-seeking experiences among infertile couples in Pakistan. Additionally, excluding male partners from the qualitative component limits insight into men's perspectives and experiences.

Additionally, our facility-based sampling excludes couples who have not yet sought medical care due to financial constraints, stigma, lack of awareness, or geographic isolation. A community-based study design would provide more comprehensive insights into barriers preventing initial healthcare contact and capture the full spectrum of infertility experiences, including those who rely exclusively on traditional or alternative remedies. Future research should employ community-based approaches, such as household surveys or population-based cohort studies, to identify and characterize infertile couples outside healthcare

Furthermore, our study did not collect detailed data on biological, clinical, or genetic factors contributing to infertility (e.g., specific diagnoses such as polycystic ovary syndrome, endometriosis, male factor infertility, or chromosomal abnormalities). We also did not separately analyze risk factors and healthcare-seeking patterns for male versus female factor infertility. This represents a significant gap, as the etiology of infertility and gender-specific attribution may substantially influence treatment-seeking decisions, stigma experiences, and emotional burden. Future studies should incorporate comprehensive clinical assessments, differentiate male and female factor causes, and examine how diagnostic outcomes influence subsequent healthcare-seeking behaviors and psychological wellbeing.

## Supporting information

**S1 Checklist. STROBE Checklist.**
(DOCX)

**S1 File. Questionnaire.**
(DOCX)

## Acknowledgments

The Authors would like to thank all the study participants for sharing their time and experiences.

## Author contributions

**Conceptualization:** Menahyl Mahmood, Babar Tasneem Shaikh.

**Data curation:** Menahyl Mahmood.

**Formal analysis:** Menahyl Mahmood.

**Investigation:** Menahyl Mahmood, Babar Tasneem Shaikh.

**Methodology:** Menahyl Mahmood, Babar Tasneem Shaikh, Mariam Ashraf.

**Resources:** Menahyl Mahmood, Babar Tasneem Shaikh, Mariam Ashraf.

**Supervision:** Mariam Ashraf.

**Validation:** Menahyl Mahmood, Babar Tasneem Shaikh, Mariam Ashraf.

**Writing – original draft:** Menahyl Mahmood, Babar Tasneem Shaikh, Mariam Ashraf.

**Writing – review & editing:** Menahyl Mahmood, Babar Tasneem Shaikh, Mariam Ashraf.

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
