## [Decision Letter · Decision Letter 0]

15 Dec 2025

PGPH-D-25-02139

Factors Influencing Infertile Couples' Decisions to Seek Healthcare: A Mixed Methods Study from Islamabad, Pakistan.

Dear Dr. Ashraf,

Thank you for submitting your manuscript to PLOS Global Public Health. After careful consideration, we feel that it has merit but does not fully meet PLOS Global Public Health’s publication criteria as it currently stands. Therefore, we invite you to submit a revised version of the manuscript that addresses the points raised during the review process.

We look forward to receiving your revised manuscript.

Kind regards,

Anushka Ataullahjan

Guest Editor

Journal Requirements:

1. Please ensure that your Ethics Statement is available in its entirety at the beginning of your Methods section, under a subheading 'Ethics Statement'.

2. Please upload separate figure files in .tif or .eps format. Also, remove the figures from your manuscript file but keep the legends.

4. We note that your Data Availability Statement is currently as follows: “all the data is already shared in the results section”

Additional Editor Comments (if provided):

Reviewers' comments:

Reviewer's Responses to Questions

**Comments to the Author**

1. Does this manuscript meet PLOS Global Public Health’s publication criteria? Is the manuscript technically sound, and do the data support the conclusions? The manuscript must describe methodologically and ethically rigorous research with conclusions that are appropriately drawn based on the data presented.? Is the manuscript technically sound, and do the data support the conclusions? The manuscript must describe methodologically and ethically rigorous research with conclusions that are appropriately drawn based on the data presented.

Reviewer #1: Yes

Reviewer #2: Partly

Reviewer #3: Partly

2. Has the statistical analysis been performed appropriately and rigorously?

Reviewer #1: Yes

Reviewer #2: Yes

Reviewer #3: No

3. Have the authors made all data underlying the findings in their manuscript fully available (please refer to the Data Availability Statement at the start of the manuscript PDF file)?

The PLOS Data policy requires authors to make all data underlying the findings described in their manuscript fully available without restriction, with rare exception. The data should be provided as part of the manuscript or its supporting information, or deposited to a public repository. For example, in addition to summary statistics, the data points behind means, medians and variance measures should be available. If there are restrictions on publicly sharing data—e.g. participant privacy or use of data from a third party—those must be specified.requires authors to make all data underlying the findings described in their manuscript fully available without restriction, with rare exception. The data should be provided as part of the manuscript or its supporting information, or deposited to a public repository. For example, in addition to summary statistics, the data points behind means, medians and variance measures should be available. If there are restrictions on publicly sharing data—e.g. participant privacy or use of data from a third party—those must be specified.

Reviewer #1: Yes

Reviewer #2: Yes

Reviewer #3: No

4. Is the manuscript presented in an intelligible fashion and written in standard English?

Reviewer #1: Yes

Reviewer #2: Yes

Reviewer #3: Yes

Reviewer #1: This study employs a mixed - methods design to explore the factors influencing healthcare - seeking decisions among infertile couples in Islamabad, Pakistan. Combining in - depth interviews and a cross - sectional survey, the research reveals significant socio - psychological pressures and economic and cultural barriers faced by these couples. The results indicate that education level, family structure, and type of infertility significantly impact their decision - making. The study highlights the need for community - based education programs, psychological counseling, and social health protection to improve healthcare - seeking behavior and quality of life for infertile couples. I have some suggestions for future improvement.

1. When discussing the cultural background of Pakistan, the text mentions “Additionally, in developing Asian countries, infertility is often seen as an act of God, a punishment for past sins, or the result of witchcraft”. Although this view is mentioned, the paper could further explore how this cultural perception specifically affects the healthcare - seeking decisions of infertile couples. For example, it could add an analysis of how this cultural belief leads to delays or resistance to medical intervention and how this belief varies among different social classes or educational levels.

2. The text states “The study population comprised married couples in Islamabad who had unsuccessfully tried to conceive for 12 to 18 months, experiencing primary and secondary infertility (having at least one live child), aged 18–35 years and were recruited from a private hospital providing infertility treatment”. The sample being drawn from only one private hospital in Islamabad may not be representative of infertile couples across different regions and economic levels in Pakistan. The authors may have had to choose a specific sample source due to resource or practical constraints. It is suggested that the limitations of the sample could be further discussed in the discussion section. For example, it could mention how this sample selection might affect the generalizability of the study results. If future studies could consider sampling from different types of medical institutions (such as public and private) and different regions (urban and rural), it would help improve the representativeness of the sample.

3. When presenting the results of the multivariate logistic regression analysis, “Regression analysis shows that individuals with primary or lower education are 0.2 times less likely to seek healthcare than those with middle or higher education”. Although the impact of education level on healthcare - seeking decisions is given, the underlying mechanisms of this impact could be further explored. For example, it could analyze, in combination with relevant literature, whether people with lower education levels have insufficient knowledge about infertility treatment or whether they face more economic pressure or other social factors.

4. In the conclusion section, it mentions “Recognising infertility as an important public health issue, it is crucial to develop policies that ensure accessible, equitable, and supportive reproductive healthcare for everyone”. Although the general direction of policy recommendations is given, more specific policy measures could be proposed in response to the specific problems found in the study, such as social stigma and economic burden. For example, it could suggest how policies could be used to change the social stigmatization of infertility through publicity or how the government could formulate subsidy policies to alleviate the economic burden on low - income families.

Reviewer #2: It would have been better if the study was conducted at community level to include all infertile couples who have not visited health care settings yet. The way the data were analyzed lacks some clarity and details. Biological, clinical and genetic factors influencing infertility were not included in the study. Risk factors for males and females were not separately identified (described).

Reviewer #3: Congratulations to the authors on the completion of the study and on putting together a good report on a relevant subject.

However, I have a few comments.

s/n Section Comments Recommendations

1 Data Collection, Management and Analysis Sampling methods seem unclear; was it systematic random as stated here or purposive sampling as stated in the limitations Consider reviewing and harmonizing these.

2. There is a repetition of the inclusion criteria in the last paragraph with additional information about use of contraception that was not part of the initial list. Consider harmonizing these and avoid repetitions

3 Quantitative findings The second sentence seeks to report the “average age” but a range was stated. Kindly state the mean (with SD).

4 The 5th sentence (Most had been married…) is unclear and has a mathematical error; 139 of 269 is not 44.6%. Consider recalculating and rewording this.

5 Table 1 Under the ‘Treatment advised by’, what does ‘None’ represent? Missing data? Considering self-seeking is a category here. Kindly provide clarifications.

6 Table 2 It is unclear how you have two categories of ‘Decision to seek medical care for infertility’ as Yes and No. Giving 100% of the participants already sought medical care.

More so, one cannot trace where the 175 participants under ‘Yes’ had come from. Kindly review and provide clarifications.

7 Similarly, there are 81 participants who were advised by family and friends on table 1 but a total of 87 on table 2 Kindly review and rework

8  Discussion Factors not adequately discussed Authors should do more in highlighting the factors identified from the study and discussing the logistic regression findings after due diligence with the analyses.

9 The sentences after the citation 29 and 30 seem unfit for that section. Consider moving those to the conclusion.

10 General comment The use of medical ‘advice’ and medical ‘treatment’ interchangeably These two are not technically the same and authors should review and decide which they are referring to.

**Do you want your identity to be public for this peer review?** For information about this choice, including consent withdrawal, please see our Privacy Policy..

Reviewer #1: **Yes:** Pengpeng YePengpeng YePengpeng YePengpeng Ye

Reviewer #2: **Yes:** Kumlachew Mergiaw AbtewKumlachew Mergiaw AbtewKumlachew Mergiaw AbtewKumlachew Mergiaw Abtew

Reviewer #3: No

---

## [Decision Letter · Decision Letter 1]

15 Mar 2026

PGPH-D-25-02139R1

Factors Influencing Infertile Couples' Decisions to Seek Healthcare: A Mixed Methods Study from Islamabad, Pakistan.

Dear Dr. Ashraf,

Thank you for submitting your manuscript to PLOS Global Public Health. After careful consideration, we feel that it has merit but does not fully meet PLOS Global Public Health’s publication criteria as it currently stands. Therefore, we invite you to submit a revised version of the manuscript that addresses the points raised during the review process.

We look forward to receiving your revised manuscript.

Kind regards,

Anushka Ataullahjan

Guest Editor

Journal Requirements:

Additional Editor Comments (if provided):

Thank you for sharing this revised manuscript, please do address a few minor comments and clarifications identified by the reviewers before resubmitting.

Reviewers' comments:

Reviewer's Responses to Questions

**Comments to the Author**

Reviewer #4: (No Response)

Reviewer #5: (No Response)

publication criteria? Is the manuscript technically sound, and do the data support the conclusions? The manuscript must describe methodologically and ethically rigorous research with conclusions that are appropriately drawn based on the data presented.? Is the manuscript technically sound, and do the data support the conclusions? The manuscript must describe methodologically and ethically rigorous research with conclusions that are appropriately drawn based on the data presented.

Reviewer #4: Yes

Reviewer #5: Yes

3. Has the statistical analysis been performed appropriately and rigorously?

Reviewer #4: Yes

Reviewer #5: No

4. Have the authors made all data underlying the findings in their manuscript fully available (please refer to the Data Availability Statement at the start of the manuscript PDF file)?

The PLOS Data policy requires authors to make all data underlying the findings described in their manuscript fully available without restriction, with rare exception. The data should be provided as part of the manuscript or its supporting information, or deposited to a public repository. For example, in addition to summary statistics, the data points behind means, medians and variance measures should be available. If there are restrictions on publicly sharing data—e.g. participant privacy or use of data from a third party—those must be specified.requires authors to make all data underlying the findings described in their manuscript fully available without restriction, with rare exception. The data should be provided as part of the manuscript or its supporting information, or deposited to a public repository. For example, in addition to summary statistics, the data points behind means, medians and variance measures should be available. If there are restrictions on publicly sharing data—e.g. participant privacy or use of data from a third party—those must be specified.

Reviewer #4: Yes

Reviewer #5: Yes

5. Is the manuscript presented in an intelligible fashion and written in standard English?

Reviewer #4: Yes

Reviewer #5: Yes

Reviewer #4: This is a well written manuscript, great flow. An important public health concern in many settings. The authors demonstrated the similarity of the issue in most settings across Asia and Africa where cultural beliefs have a great influence on health-seeking behavior and social norms.

It could be easier for the reviewer to comment and for the author to track comments if the manuscript lines are numbered

When authors determine the required sample size, it is appropriate to report the actual size used if it differs from the required

The description of the qualitative analysis is insufficient. The authors state that thematic content analysis was conducted manually, but additional details are needed regarding the analytic approach, coding procedures, number of coders, development of the codebook, consensus procedures, saturation assessment.

“Data triangulation was employed to ensure validity and reliability”. Kindly give details on this process.

In the results table 1 Factors influencing the decision to seek treatment, the authors reported 4 items, one of which is unable to conceive, which is the definition of infertility. This mimics the title of the table, not results, as the other 3 items are all describing the reason to seek treatment for the inability to conceive. This item doesn’t give accurate information.

In the discussion, the association between social factors and lower educational attainment may not consistently predict health behaviors. Highly educated and well-informed individuals often adhere strongly to generational beliefs and cultural norms that influence health-related perceptions and opinions

Reviewer #5: The study would benefit from data from study location on the infertility rates and then presented in the whole Country perspective. In the research questions, not sure why the first 2 questions were asked on the couples and the third yet important one only on women? Do we have findings that support that challenges are faced more or only by women? As for mixed methods study, as quantitative surveys were filled by couples and women who ere seeking treatment in one private healthcare facility, with qualitative interviews, why only woman were sampled? It is imperative that couples are sampled in the design to justify the research objective.

Logistic regression findings on the predictors (educational level, family structure, type of infertility) look like could be chance finding. Would like to get the statistical teams comment on this.

The recommended policy interventions need feasibility and pragmatic approach. As a suggestion would be good to have Key Informant Interviews (KIIs) with the key stakeholders on this agenda and plan what is appropriate.

The limitation on the sampling from one private hospital will need more attention as the recommendations made could be affected or might have to be varied based on the sampling location and sampling characteristics. The recommendation to run a community level research exploring male factors will need followup and strategies to make it a reality. The engagement of key stakeholders on the current recommendations needs addressing and planning to make sustainable and meaningful policy level interventions.

As for the study design, the questionnaire used looks like the responses could have been influenced by the nature of the questions. The responses on sensitive areas might not yield valid responses if asked as binary or leading questions. The strong suggestion would be to use qualitative study (KIIs with couples, Focus group discussions (FGDs) with families, community, health care professionals) to get comprehensive narration of the underlying contributing factors and the intrinsic mechanisms that operate and influence the healthcare seeking behaviour of the sample studied.

**Do you want your identity to be public for this peer review?** For information about this choice, including consent withdrawal, please see our Privacy Policy..

Reviewer #4: **Yes:** MARIA AFADAPAMARIA AFADAPAMARIA AFADAPAMARIA AFADAPA

Reviewer #5: **Yes:** Somasundari GopalakrishnanSomasundari GopalakrishnanSomasundari GopalakrishnanSomasundari Gopalakrishnan

---

## [Editor Report · Decision Letter 2]

24 Mar 2026

Factors Influencing Infertile Couples' Decisions to Seek Healthcare: A Mixed Methods Study from Islamabad, Pakistan.

PGPH-D-25-02139R2

Dear Dr Ashraf,

We are pleased to inform you that your manuscript 'Factors Influencing Infertile Couples' Decisions to Seek Healthcare: A Mixed Methods Study from Islamabad, Pakistan.' has been provisionally accepted for publication in PLOS Global Public Health.

Best regards,

Anushka Ataullahjan

Guest Editor